# Acute Promyelocytic Leukemia: Update on the Mechanisms of Leukemogenesis, Resistance and on Innovative Treatment Strategies

**DOI:** 10.3390/cancers11101591

**Published:** 2019-10-18

**Authors:** N. I. Noguera, G. Catalano, C. Banella, M. Divona, I. Faraoni, T. Ottone, W. Arcese, M. T. Voso

**Affiliations:** 1Department of Biomedicine and Prevention, Tor Vergata University of Rome, 00133 Rome, Italy; gianfranco.catalano@uniroma2.it (G.C.); cristina.banella@gmail.com (C.B.); tiziana.ottone@uniroma2.it (T.O.); william.arcese@ptvonline.it (W.A.); Voso@med.uniroma2.it (M.T.V.); 2Santa Lucia Foundation, Unit of Neuro-Oncoematologia, Istituti di Ricovero e Cura a Carattere Scientifico (IRCCS), 00143 Rome, Italy; 3Policlinico Tor vergata, 00133 Rome, Italy; mariadomenica.divona@ptvonline.it; 4Department of Systems Medicine, University of Rome Tor Vergata, 00133 Rome, Italy; faraoni@med.uniroma2.it

**Keywords:** APL, therapy, NGS, resistance, ATO, ATRA, ascorbate

## Abstract

This review highlights new findings that have deepened our understanding of the mechanisms of leukemogenesis, therapy and resistance in acute promyelocytic leukemia (APL). Promyelocytic leukemia-retinoic acid receptor α (PML-RARa) sets the cellular landscape of acute promyelocytic leukemia (APL) by repressing the transcription of RARa target genes and disrupting PML-NBs. The RAR receptors control the homeostasis of tissue growth, modeling and regeneration, and PML-NBs are involved in self-renewal of normal and cancer stem cells, DNA damage response, senescence and stress response. The additional somatic mutations in APL mainly involve FLT3, WT1, NRAS, KRAS, ARID1B and ARID1A genes. The treatment outcomes in patients with newly diagnosed APL improved dramatically since the advent of all-trans retinoic acid (ATRA) and arsenic trioxide (ATO). ATRA activates the transcription of blocked genes and degrades PML-RARα, while ATO degrades PML-RARa by promoting apoptosis and has a pro-oxidant effect. The resistance to ATRA and ATO may derive from the mutations in the RARa ligand binding domain (LBD) and in the PML-B2 domain of PML-RARa, but such mutations cannot explain the majority of resistances experienced in the clinic, globally accounting for 5–10% of cases. Several studies are ongoing to unravel clonal evolution and resistance, suggesting the therapeutic potential of new retinoid molecules and combinatorial treatments of ATRA or ATO with different drugs acting through alternative mechanisms of action, which may lead to synergistic effects on growth control or the induction of apoptosis in APL cells.

## 1. Introduction

Acute promyelocytic leukemia (APL) is a subtype of acute myeloid leukemia (AML) which is characterized by a reciprocal and balanced translocation involving the retinoic acid receptor α (*RARa*) on chromosome 17 and promyelocytic leukemia (PML) gene on chromosome 15, and generates the oncogenic PML-RARα fusion protein [1,2,3]. PML-RARα has two primary effects: deregulates transcriptional control (acting as a transcriptional repressor of RARα target genes) and disrupts PML homeostatic function. RARα is a ligand-dependent transcription factor that binds to retinoid X receptors (RXR) to form transcriptionally active heterodimers. The retinoic acid (RA) response elements (RARE) are found in genes that play pivotal roles in a series of physiological processes including cell growth, differentiation, survival and death. Retinoid receptors also activate kinase signaling pathways, which fine-tune the transcription of the retinoic acid target genes [4]. The PML gene, due to alternative splicing of its C-terminal exons, displays six nuclear and one cytoplasmic isoforms. The PML protein plays a role in a number of cellular processes involving homeostasis and tumor suppression mechanisms, including the response to viral infections and stress, senescence, angiogenesis, differentiation, and maintenance of genome stability. To exert its functions, PML recruits a molecular apparatus organizing peculiar organelles known as nuclear bodies (PML-NBs) [5,6]. The PML-RARα fusion protein acts as a transcriptional repressor of RARa target genes leading to the proliferation of myeloid progenitors and maturation arrest at the promyelocytic stage, and disrupts PML-NBs [7,8,9,10]. The present review highlights new findings that have deepened the understanding of the mechanisms of leukemogenesis, and resistance, enabling the design of experimental treatment strategies in APL.

## 2. APL Pathophysiology

RAR receptors (alpha, beta and gamma) signaling is pivotal in the homeostatic control of tissue growth, modeling, and regeneration. The evidence suggests that the RAR transcriptional pathway regulates hematopoietic stem cell (HSC) development, maintenance and expansion, as well as maturation/differentiation into distinct hematopoietic cell lineages.

Ligand-free RARs bind to RAREs on the DNA to repress the transcription of RAR responsive genes through the assemblage of the RAR co-repressors. These include NR corepressor (NCoR) and SMRT which trigger the recruitment of a high molecular weight complex containing histone deacetylase (HDAC) to maintain chromatin in a densely packed, inactive state. Retinoid binding induces conformational changes which cause the dissociation of co-repressors and the induction of transcriptional activation of responsive genes [4]. PML-RARα takes control of RARE sites and has a dominant negative action on transcription, inhibiting activation by physiological ligands, causing the maturation arrest at the promyelocyte stage.

PML NBs dysfunction has been implicated in various cellular processes leading to the APL oncogenic phenotype [7,11]. Senescence is known to be the first physiological defense against cellular transformation. Two key observations have directly implicated PML in senescence: Ras-induced senescence is lost in a pml−/− context, conversely PML overexpression induces premature senescence [12,13]. PML NBs are stress-sensitive and are required for p53 action: Cells lacking PML show a reduced propensity to undergo senescence or apoptosis in response to p53 activation [14,15]. Notably, two of the most active drugs in APL therapy, ATRA and ATO, allow the reformation of PML NBs as the result of PML-RARα degradation [16,17].

PML-NBs also have an important role in the mechanisms of DNA damage response (DDR) via both NHEJ and HR repair pathways. Its disruption by PML-RARα strongly affects ATM activation, as well as CHK2 and NBN phosphorylation [18,19]. ATM kinases are clients of the HSP90α chaperone, whose inhibition leads to the destabilization of these important components of the DNA damage response [20]. Recently, the authors demonstrated that Hsp90 is downregulated in the presence of PML-RARα [21], thus could hamper the DNA damage response in APL cells.

PML NBs are involved in the self-renewal of normal and cancer stem cells [22,23]. PML contributes to embryonic stem cells (ESC) self-renewal maintenance by controlling cell-cycle progression and sustaining the expression of crucial pluripotency factors [5]. Among these, phosphatase and Tensin homologue deleted on chromosome 10 (PTEN) is an important tumor suppressor and plays a pivotal role in the self-renewal of hematopoietic stem cells. Its ablation promotes the exhaustion of normal hematopoietic stem cells (HSCs) and the generation of leukemia-initiating cells (LICs) [24,25]. Although the ubiquitinylation and subcellular localization of PTEN are regulated by a HAUSP-PML network [26], the authors demonstrated that PML-RARα directly suppresses PTEN expression [27]. The lack of PTEN anticancer control could play a pivotal role in favoring the emergence, survival and proliferation of damaged stem cells, allowing the accumulation of additional genetic events towards leukemogenesis.

The PML-RARα phenotype is also characterized by the downregulation of autophagy-related genes (ATGs) (i.e., ULK1, BECN1, ATG14, ATG5, ATG7, ATG3, ATG4B and ATG4C) [28,29,30]. Autophagy or self-eating is a highly conserved, closely regulated homeostatic cellular activity that allows for the bulk degradation of long-lived proteins and cytoplasmic organelles [31]. Autophagy is one of the main cellular catabolic pathways controlling a variety of physiological processes and its disruption interferes with the machinery of self-renewal, differentiation and death [32]. Its roles in cancer initiation and progression and in determining the response of tumor cells to anticancer therapy are complicated. A recent study provided evidence for a new function for PML as a repressor of autophagy when associated at the mitochondrial-associated membrane (MAM) [33]. However, whether or not PML-RARa inhibits autophagy through its binding to MAM has not yet been investigated. These observations differ from the data reporting that PML-RARa is involved in constitutive activation of autophagy through inhibiting the AKT/mTOR pathway in APL cells [34]. The reason for seemingly opposite effects of PML-RARa on the regulation of autophagy remains to be clarified. Figure 1 depicts the pathophysiologic aspects of APL (Figure 1).

Ethnicity is considered a measure of genetic and lifestyle. Some studies, mainly from Latino America, suggested that the Latino population had a higher proportion of APL among all AML diagnosis, which ranged approximately 30–38% against less than 10–7% in the non-Latino population [35,36,37] Nevertheless, Latino is not an ethnicity and it is very difficult to define the characteristics of this population. On the other hand, several reports did not find any significant variation in the incidence, and there is no hard evidence of any ethnic physiological or life style difference linked to APL incidence.

## 3. Additional Genetic Events

The PML-RARα rearrangement is the main pathogenetic event of APL. However, its expression in the bone marrow in a mouse model leads to a myeloproliferative phenotype similar to chronic myeloid leukemia, while the emergence of full-blown APL requires 12–14 months [12,38]. This long latency suggests that additional genetic/epigenetic changes are necessary to flare the APL phenotype. By next generation sequencing (NGS), a recurring point mutation (*Jak1* V657F or V658F in humans) and a recurring deletion in histone demethylase *Kdm6a* were identified in a mouse APL model [39]. In vivo data have shown that aberrant Jak1 signaling can play an important role in APL pathogenesis and that the JAK-STAT signaling may be activated by different mechanisms in a mouse model of APL. The role of *KDM6A* mutation in APL pathogenesis is not clear yet, but it may involve the dysregulation of Hox, Notch, or Rb signaling pathways [40,41,42,43].

The topography of somatic mutations in APL is defined by recurrent alterations of FLT3, WT1, NRAS, KRAS, ARID1B and ARID1A genes, and the near-absence of mutations in other common AML genes [44,45,46,47]. A recent survey of 153 primary and 69 relapse APL samples by whole-genome sequencing showed that both the primary and relapse APL harbored an average of eight non-silent somatic mutations per exome. The recurrent alterations of *FLT3* (43%), *WT1* (14%), *NRAS* (10%) and *KRAS* (4%) in newly diagnosed APL were reported, whereas mutations in other genes commonly mutated in myeloid leukemia including *DNMT3A*, *NPM1*, *TET2*, *ASXL1* and *IDH1/2* were absent [44]. In a Chinese report on 84 APL samples studied by RT-PCR, the prevalence of mutations was 60.7% (51/84), with 27.4% *FLT3-ITD* mutations, 14% *WT1* mutation, 10% *FLT3-TKD*, 8% *TET2*, 6% *N-RAS*, 5% *ASXL1*, 2% *EZH2* mutations and 1% each for *MLL-PTD*, *IDH1* and *CBL* mutations, respectively. No mutations were found in other common leukemia-related genes: *JAK1*, *DNMT3*, *c-Kit*, *NPM1*, *IDH2*, *RUNX1* and *JAK2* (V617F) [45]. The authors reported a low frequency of mutations of *WT1* exon 7 and 9 in 103 APL samples (4%) [48]. Activating *FLT3* signaling mutations are often present in APL, associated with hyperleukocytosis, a major adverse risk factor in chemotherapy-based regimens [49,50]. FLT3, KRAS, trisomy 8 and the deregulated expression of BCL2 have been described as cooperators of PML-RARa in mouse models of APL [51,52]. Recently, Esnault et al. demonstrated, also in an APL mouse model, that *FLT3-ITD* mutations severely blunt ATRA response. The combination of ATRA and ATO fully rescued the therapeutic response in *FLT3-ITD+* APLs, leading to PML-RARa degradation, PML nuclear body reformation, P53 activation and APL eradication [53]. This review further reinforces the biological importance of the presence of *FLT3-ITD* mutations independent from its allelic ratio in the context of the disease. A recent report by Iaccarino et al. described a median of two additional mutations per patient (range, 0–3) at the time of APL diagnosis, in 11 patients who later relapsed and a median of one mutation (range 0–2) in 21 of 33 APL patients, who remained in continuous complete remission (*P* = 0.0032). They also found NRAS and RUNX1 mutations only in patients who later relapsed, with *FLT3* (ITD/TKD: 5 of 11 patients, 45%) and NRAS (3 of 11 patients, 27%) as the most frequently mutated genes. The *FLT3-ITD* mutations were present in most high-risk patients, but were not significantly associated with the risk of relapse. The authors suggest that molecular patterns detectable at diagnosis may predict the treatment response. Comparing the initial mutational status of patients who relapsed during the course of disease (*n* = 11) with those in CCR (*n* = 33), they found differences in concomitant mutations, with *NRAS* and RUNX1 mutated only in the first group of patients, suggesting their possible role as predictive markers of relapse. Of note, also the number of concomitant mutations per patient was significantly higher in the group who later relapsed (*P* < 0.0001) indicating an accumulation of genetic alterations during disease progression. In particular, they found mutations associated with clonal hematopoietic expansion, like *ASXL1*, *DNMT3A*, *JAK2*, *SRSF2*, *TET2* and *TP53* [46]. However, validation studies are needed to confirm these hypotheses.

The optimal management of APL requires the early diagnosis, institution of aggressive supportive measures, appropriate management of treatment-related complications, and monitoring of measurable residual disease (MRD) for the presence of PML/RARa. MRD monitoring is not recommended in non–high-risk patients who achieve profound molecular remission (CRMRD2) status after consolidation, independently from induction treatment (ATRA plus ATO or ATRA plus chemotherapy) [54]. Contrarily, high-risk patients and those not responding well to induction therapy request frequent and accurate monitoring. As mentioned before, a relatively large set of genes are mutated or aberrantly expressed in APL. These genetic alterations, over their significance in prognosis, might be of use as clonal markers in measuring MRD after treatment. The availability of new more sensitive methods of investigations, like deep sequencing and mutation-specific ddPCR, enable a better understanding of the clonal evolution in cases harbouring several mutations acquired during disease progression. However, apart from high-risk characteristics, the question remains on how to discriminate at onset the cases that need a more-timely diagnostic approach. The first study to propose a molecular risk score in APL was reported by Hecht et al. Although the sample size was limited (79 patients), Hecht et al. demonstrated that *BAALC*, *ERG* and *WT1* expression levels integrated into a score that could offer a promising approach to guide the monitoring of patients with APL treated with ATRA and high doses of cytarabine [55,56]. Recently, Lucena-Araujo et al. in a recent study, proposed the use of an integrative score in APL (ISAPL) based on *FLT3-ITD* mutational status, ΔNp73/TAp73 expression ratio, *ID1*, *BAALC*, *ERG* and *KMT2E* gene expression levels. They proposed that the combination of gene mutation with gene expression analysis could improve the outcome prediction in acute promyelocytic leukemia. Lucena-Araujo et al. suggested that a better understanding of the molecular limitation of patients in response to ATRA and anthracycline-based chemotherapy could serve as the basis for future initiatives aiming to change the current scenario for APL treatment in patients who live in low- and middle-income countries (LMIC) [57]. The additional mutations events are resumed in Table 1.

## 4. Immunophenotypic Characteristics

The APL cells show absent or minimally expression gp-170 as well as other proteins associated to multidrug resistance, such as MRP1, MRP2, and LRP, present in a high proportion of AMLs. This is a feature that may be relevant to explain the striking sensitivity of APL blasts to anthracyclines [58]. APL is characterized by low or negative CD34 expression, the infrequent expression of HLA-DR and the lack of CD7, CD11a, CD11b, CD14 and CD18, strong positivity for CD33, the expression of CD13 and CD117 [59], and frequently show aberrant expressions of the T-cell-associated antigen CD2 which is associated with the microgranular variant morphology and increased leukocyte counts at presentation [17,58,60]. CD56 expression has been associated with resistance to standard ATRA and chemotherapy [61]. In a recent Chinese review of 798 APL cases, the authors in that study described high SSC, the absence of expression of CD34 and HLA-DR, the strong expression of CD33 in 90% of cases, the consistent expression of CD13, CD9, CD123, and the expression of CD56, CD7, CD2 (sometimes). The remaining 10% of the cases showed atypical APL phenotypes, positive for CD34 and/or HLA-DR expression, with decreased SSC and a frequent CD2 expression [62]. This suggests that an inclusive approach is required to indicate further genetics or molecular biology tests at diagnosis.

## 5. Insight into the Mechanisms of Treatment Resistance in APL

The knowledge on the mechanisms of resistance to treatment in APL is vitally important to develop new therapies. ATRA has a double therapeutic function in APL: activates the transcription of genes involved in myeloid lineage differentiation and degrades the PML-RARα oncoprotein. ATO degrades all PML containing molecular species, promoting apoptosis in APL cells, has a pro oxidant effect and damages protein structures at large by resolving disulphide bonds. Resistance to ATRA and ATO could derive mechanically from genetic mutations resulting in amino acid substitution in the RARa ligand binding domain (LBD) and in the PML-B2 domain of PML-RARa respectively [46,63,64,65]. LBD mutations are confined to three clusters regions and were confirmed in 18 out of 45 (40%) relapsed patients treated with ATRA-chemotherapy [64]. Dicysteine C212/C213 in PMLB2 domain is critical for direct ATO binding and for the serial reaction of sumoylation, multimerization, and degradation of PML-RARa [66]. However, such mutations cannot explain the majority of resistance to ATO experienced in the clinic.

Recently Lehmann-Che et al. studied 64 matched samples collected from patients at initial diagnosis, during remission, and following relapse after ATRA-chemotherapy treatment by exome sequencing. They confirmed that APL is a relatively simple disease under the genetic profile which at diagnosis is similar in patients who later relapse or remain in complete remission, except for a significant excess of WT1 mutations or loss (7/18, 40%) in relapsing patients, and, importantly, that some relapses were completely distinct from the diagnostic APL clone [65]. This has been previously observed also in chemotherapy-treated core-binding factor leukemias [67]. Lehmann-Che, et al. identified, in addition to WT1 mutations, rare anomalies involving activators of MAP kinase pathway and/or other epigenetic controllers that disrupt key epigenetic or transcriptional regulators. Once relapsed, the disease often acquires additional oncogenic alterations and/or mutations impairing the treatment response (RARa, NT5C2). Some mutations present at diagnosis were lost upon relapse, mainly including FLT3 and other passenger mutations. The data infer that the relapses derive from PML-RARA expressing clones, different from the one expanded at disease onset, that survived ATRA-chemotherapy [65].

By NGS, a 31 myeloid gene panel in 33 patients in continuous complete remission (CCR) and in 11 relapsed APL patients were analyzed, including four cases with multiple relapses. All APL with multiple relapses after ATRA-ATO displayed a significantly higher number of mutations as compared to CCR patient samples, indicating an accumulation of genetic alterations during disease progression. In particular, the authors found mutations associated with clonal hematopoietic expansion, like *ASXL1*, *DNMT3A*, *JAK2*, *SRSF2*, *TET2* and *TP53* [46]. The mutational patterns suggested different models of disease progression. In some patients, relapses may originate from the driver clone present at diagnosis. In other cases, relapses probably emerged from ATO or ATRA-resistant subclones. Particularly, PML-mutated subclones seemed to arise under the selective pressure of ATO treatment.

Esnault et al. reported that FLT3-ITD severely blunts ATRA response, not preventing PML-RARα dislocation of NCoR/HDAC complex from the RARE motifs, but protecting the oncoprotein from degradation. Thus, although in the presence of FLT3 mutations the transcriptional output of the initial ATRA response is unaffected, the retinoid fails to degrade PML-RARA protein, whose persistence in the cells confers the APL phenotype with PML nuclear body disruption and the deactivation of P53 signaling. The resistance is overcome by ATO [53], reflecting the efficacy of the ATRA/ATO combination in patients with FLT3 mutation proven in different clinical studies [68,69].

Autophagy is another mechanism involved in APL resistance. The relationship between autophagy and cancer is complex, with both deficiency and over-activation contributing to the development and progression of cancer. In general, during the advanced stages of the malignant disease, clonal evolution reactivates autophagy to provide for the metabolic needs of the cells and to promote tumor growth, invasion and metastasis. The APL cells display low levels of expression of autophagy genes and reduced autophagy activity: ATRA restores autophagy in these cells allowing granulocyte differentiation through the degradation of PML-RARa [70]. ATO also promotes autophagy-dependent clearance of PML-RARa in APL cells [70,71,72]. Hence, enhancing autophagy may have therapeutic benefits in maturation-resistant APL cells. However, the role of autophagy following APL therapy is not so simple, since some autophagy regulatory proteins (e.g., BECN1 and p62/SQSTM1) have been shown to play a pro-survival role during ATRA and ATO treatment. This might contribute to the development of resistance to treatment [32].

Alex et al. used whole exome sequencing to address ATO-resistance mechanisms in three NB4 APL cell line clones developed under ATO treatment pressure. They identified alterations in the redox system, the ubiquitin-proteasome degradation pathway and the PI3-AKT signaling pathway [73]. Consistent with this report, low levels of ROS, glutathione and glucose uptake have been observed in ATO-resistant NB4 cells, proposing a metabolic rewiring hypothesis as ATO-resistance mechanism in APL cells [74]. Chendamarai using micro-array expression profiling studied 8 ATO-sensitive and 8 ATO-resistant patient samples, and found the differential regulation of the following functional pathways: (i) Cell adhesion: Integrins, Cadherins and Mucins; (ii) cell survival and anti-apoptosis: PI3-AKT, PTEN, NFĸB, MAPK and JAK-STAT; (iii) stem cell regulation: Wnt, Hedgehog and CD34; (iv) immune regulation: TNF-receptor super family genes, Interleukins. The multi drug resistant (MDR) glycoprotein is scarcely expressed by APL cells [75]. There is evidence that after relapse and in therapy resistant NB4 cell subclones, MDR and other detoxification related proteins (as MRP1) seem to be upregulated, but the role of detoxification is not capital in the resistance mechanism. P-glycoprotein (P-gp) and multidrug resistance-associated protein 1 (MRP1) are induced by arsenic trioxide (As2O3), but are not the main mechanism of As2O3-resistance in acute promyelocytic leukemia cells [76].

However, very little is known on the role of the microenvironment in APL. Several studies have shown that RARs regulate non-hematopoietic cells present in the bone marrow microenvironment influencing the stem cell fate [77,78,79].

Su M. et al. proceeded from the premise that AML cells are responsive to ATRA in vitro, whereas the drug is not efficient in the clinic, and that APL patients relapse after ATRA monotherapy. Su M. et al. hypothesized that the enhanced activity of ATRA metabolizing enzyme cytochrome P450 gene CYP26 by stromal cells could contribute to the rescue of APL cells and to the persistence of the residual disease [80].

The PML function is essential for immunosurveillance and has been proven that its degradation via ubiquitination promotes lung cancer progression by fostering an immunosuppressive and prometastatic tumor microenvironment [81]. During therapy, the degradation of PML by ATO also occurs in the BM stromal cells mimicking this effect, and contributes to sparing APL stem cells in relapsing patients. A recent report provided evidence for the involvement of microenvironment-mediated drug-resistance in ATO treated APL cells, which is driven by the nuclear factor kappa B (NF-kappa B) pathway [82].

Finally, a very limited number of patients showing an APL phenotype devoid of the t(15;17), exhibited a variety of X-RARa fusions: PLZF [83]; NuMA [84], NPM [83], STAT5b [85], FIP1L1 [86], PRKAR1A [87], ZBTB16 [88]; BCoR [89]; OBFC2A [90]; TBLR1 [91]; GTF2I [92]; IRF2BP2 [93] and FNDC3B [94]. Interestingly, most patients expressing X-RARa show clinical resistance to ATRA and/or ATO, but the molecular mechanism involved is not well known. In the case of PLZF-RARa, the resistance to ATRA has been associated with the reciprocal transcript RARa-PLZF via CRABPI-upregulation [95]. The molecular mechanisms of ATRA resistance in STAT5b-RARa could be related to aberrant transcription regulation of STAT5b target genes, but the mechanism must be investigated. Further, ZBTB16 and TBLR1 are also ATRA-resistant. ATO resistance in clinical setting has been observed in patients expressing PLZF-, STAT5b-, BCoR- and ZBTB16-RARa [88]. The lack of ATO binding sites in X-RARa proteins may cause the resistance to ATO treatment (Figure 2).

## 6. Experimental Strategies for the Treatment of Resistant APL

The treatment outcomes in patients with newly diagnosed APL have improved dramatically in the last three decades since the advent of ATRA. Some years later, the introduction of ATO-ATRA combined with chemotherapy resulted in cure rates above 80%, but was associated with the risk of severe infections and secondary leukemias [96,97,98,99]. ATO has been shown to act synergistically with ATRA to induce the degradation of the PML-RARA oncoprotein [100] and the chemo-free ATRA-ATO approach is nowadays regarded as the first treatment choice for patients with non-high-risk APL [68,69,101]. A recent review from the European leukemia network details the guidelines in the management of frontline and relapsed APL and the specific recommendations for the identification and management of the most important complications: bleeding disorder, differentiation syndrome, QT prolongation and all the other toxicities related to treatment with ATRA and ATO [54].

When diagnosed and treated promptly, APL is curable in the vast majority of patients, yet approximately 5% of cases are resistant to standard therapy and 5 to 10% relapse and eventually become resistant. In these patients, hematologic stem cell transplantation (allo-HSCT) is the only curative approach [54]. ATRA is the retinoid of choice, but other retinoids are being developed. A recent randomized study of the Japanese Adult Leukemia Study Group comparing the new retinoid molecule tamibarotene demonstrated a significant relapse-free survival benefit over ATRA as maintenance therapy [102].

Considerable effort has been dedicated to identify specific targets and novel compounds acting in synergy with, and preventing resistance to retinoids and ATO. The HDAC inhibitors as sodium butyrate, valproic acid and trichostatin A have been used in combination with ATRA in the attempt to inhibit co-repressor complexes that contain HDACs recruited by PML-RARα [103]. Gemtuzumab ozogamicin (GO), an anti-CD33 monoclonal antibody linked with calicheamicin, efficiently targets highly CD33- expressing APL cells [104,105,106].

The authors recently reported that megadoses ascorbate (ASC) induces the degradation of PML-RARa and causes apoptosis in vitro in a variety of human myeloid cell lines, including ATRA-and ATO-resistant cell lines, and cord blood-derived normal CD34+cells [107,108].

ATO itself is a pro-oxidant factor downregulating ROS scavenging proteins and disrupting redox pathways [66,109,110], which may act synergistically with ASC at high doses, also a potent oxidant [111,112,113]. Indeed, the blasts from APL patients were highly sensitive to the ASC-ATO combination including a PLZF-RARa positive case. The APL cells higher sensitivity to the redox unbalance is due in part to PML-RARα interfering with the NRF2 subcellular distribution and inhibiting its transcriptional activity (manuscript in preparation). Of note, ASC treatment causes inhibition of the activated FLT3 signaling [107] and the combination with ATRA can be of help in FLT3-mutated cases, since FLT3-ITD mutation blocks the therapeutic response to ATRA in an APL mouse model [53]. Recently, Masciarelli et al. observed a strong synergistic cytotoxic effect of ATO and the endoplasmic reticulum (ER) stress-inducing drug Tunicamycin (Tm) in ATRA sensitive and resistant APL cell lines [114]. Gu et al. reported that pharicin B, a novel natural entkaurene diterpenoid and, a family of compounds with a long-standing history of medical applications as traditional Eastern remedies, stabilizes RARα protein and has a synergistic effect with ATRA in inducing differentiation in AML cells. It also overcame ATRA resistance in two NB4 subclones [115]. Wang et al. identified a novel synthetic small compound, named LG-362B, that induces caspases-mediated degradation of PML-RARα, overcoming ATRA resistance and inducing cellular differentiation in transplantable APL murine models [116]. Calvo et al. reported that Benznidazole (BZL), a nitro aromatic anti-parassitary drug, inhibits the proliferation of leukemic cells by blocking the cell cycle at the G0/G1 phase through the up-regulation of p27 [117]. Ying et al reported that 2-bromopalmitate, an inhibitor of fatty acid oxidation, overcomes ATRA resistance in cell lines and in vivo APL mouse models. Mechanistically, 2- Bromopalmitate covalently binds and stabilizes RARα protein, leading to the enhanced transcription of RARα-target genes. Lu et al. proposed the combination of ATRA and 2-Bromopalmitate as a promising therapeutic strategy to overcome resistance in relapsed APL patients [118]. Ganesan et al. reported a synergistic effect of ATO plus Bortezomib in both ATO-sensitive and resistant APL cell lines. The mechanism of the synergy involved the downregulation of the NFĸB pathway, and an increase in the unfolded protein response (UPR) and in reactive oxygen species generation in the malignant cell. PML-RARa oncoprotein is effectively cleared with this combination despite the inhibition of the proteasome by bortezomib, since the clearance is mediated through a p62-dependent autophagy pathway [82]. As mentioned before, autophagy downregulation by PML-RARα contributes to the differentiation block, therefore potentiation of this mechanism could be a desired strategy for the differentiation therapy of APL. In this line, rapamycin and lithium, two well-known activators of autophagy, may enhance the therapeutic effectiveness of both ATRA and ATO in APL cells. Recently, Hussain et al. reported that phenylarsine oxide (PAO), one of the organic arsenic derivatives, could induce PLZF-RARα degradation through the ubiquitin proteasome degradation pathway and cause apoptosis [119]. The experimental strategies for resistance are listed in Table 2.

## 7. Prophylaxis for Incidence of CNS Relapse

In the ATO era, central nervous system (CNS) involvement is rare in APL, but causes poor prognosis. There are no hard biological data available concerning APL CNS localization, although it has been associated with several factors including high WBC count (>10 × 10^9^/L), CNS hemorrhage, expression of CD2, and/or CD56 in blasts, PML-RARA bcr3 isoform, differentiation syndrome, and above all induction therapy with single agent ATRA and regimens without Cytarabine [120,121,122]. A possible association between the use of ATRA and the development of extramedullary disease was disproved [123]. However, it is conceivable that the emergence of such a complication was more apparent due to better therapy and the prolonged survival of resistant patients. ATO is known to cross the blood-brain barrier and has CNS penetration at therapeutically meaningful levels (CSF concentration at 20% to 50% of plasma concentration). Recently, Sanz et al. reported that there are no formal data supporting the use of CNS prophylaxis in the ATO era [54]. Furthermore, Larson et al. did not recommend intrathecal chemotherapy (ITT) in the treatment of APL for any risk, thus prophylactic ITT was not used in trials that incorporated ATRA and ATO [69,124,125,126].

## 8. Conclusions and Future Perspectives

Since the first insights into the PML-RARa network and the differentiating effect of retinoids, APL has been a fascinating field for researchers as a first example of an acute myeloid leukemia cured without antiblastic chemotherapy. In all, a very successful history of setting the clinical standards, at least in countries without economical restraints for the expenses necessary for early diagnosis and treatment. One of the main challenges in APL to date is prompt diagnosis and treament, since the early death rate still reaches 15%, which is unacceptable in this disease. Yet, approximately 5 to 10% of patients will relapse and may become resistant. Thus, identification of patients at risk of clonal evolution and resistance is of vital importance. The complete definition of the genomic landscape of APL and the characterization of meaningful accessory genetic alterations, point mutations, gene copy number amplification or loss, and modification of the epigenome may pave the way for a deeper understanding of the regulatory networks in APL cells. In particular, definition of metabolic peculiarities may provide useful prognostic information and therapeutic targets, facilitating the development of active and safe agents to be tested in clinical trials.

## Reference

## Figures and Tables

**Figure 1 cancers-11-01591-f001:**
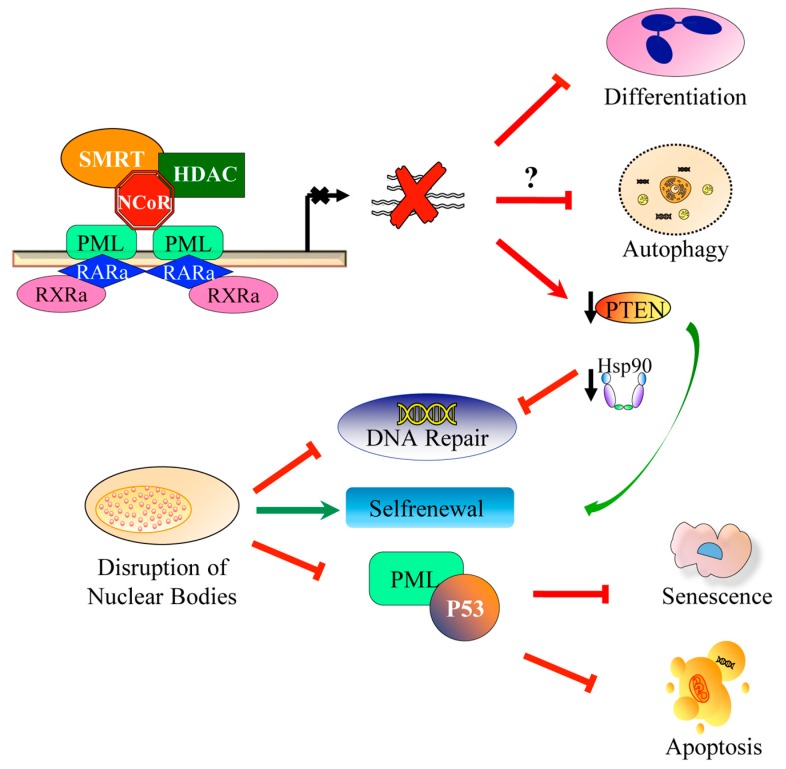
Schematic representation of the molecular mechanisms involved in acute promyelocytic leukemia (APL) pathogenesis. Promyelocytic leukemia / retinoic acid receptor α (PML/RARA) exerts dominant-negative effects on RAR/RXR-dependent transcriptional control through the recruitment of co-repressor complexes (CoR) (top) and PML nuclear bodies assembly (bottom). The direct or indirect regulation of target genes is responsible for the differentiation block, aberrant self-renewal, and impairment of autophagy and apoptosis observed in APL blasts. PML nuclear bodies disruption drives enhanced self-renewal, inhibition of DNA damage response and inhibition of senescence and apoptosis, in part by p53 inactivation.

**Figure 2 cancers-11-01591-f002:**
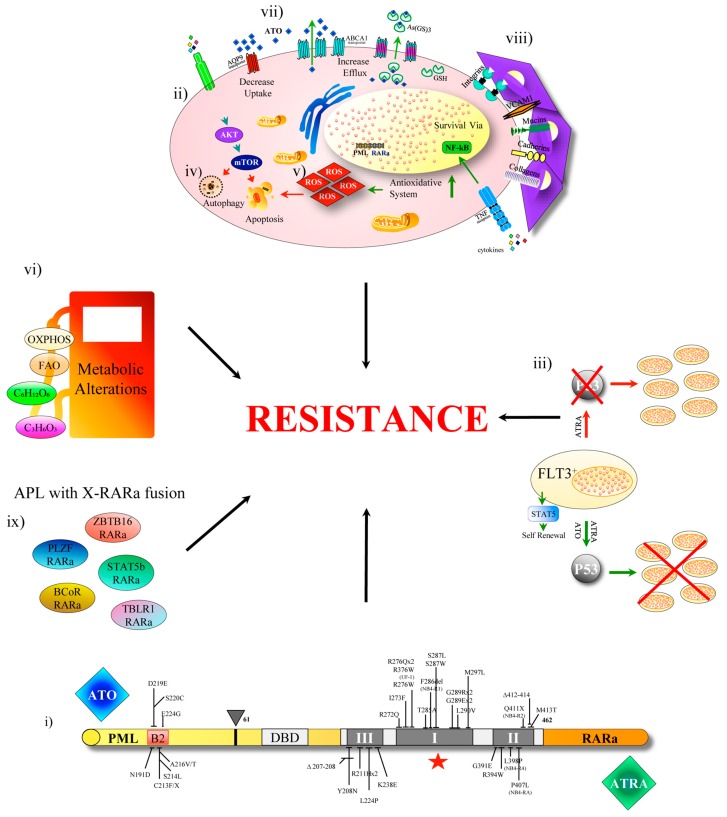
Mechanism of resistance to arsenic trioxide- all-trans retinoic acid (ATO-ATRA) therapy in APL. The resistance to ATRA and ATO may derive from: (i) genetic mutations resulting in amino acid substitution in the RARa ligand binding domain (LBD) and in the PML-B2 domain of PML-RARa (ii) deregulated pathways like AKT/mTOR, activators of MAP kinase pathway and/or other epigenetic controllers or additional gene mutations (ei *WT1*), (iii) FLT3-ITD severely blunts ATRA response, which fails to degrade PML-RARA protein whose persistence in the cells confers the APL phenotype with PML nuclear body disruption and deactivation of P53 signaling. This type of resistance is overcome by ATO (iv) some autophagy regulatory proteins BECN1 and p62/SQSTM1 have been shown to play a pro-survival role during ATRA and ATO treatment, (v) alterations in the redox system. (vi) Metabolic alterations (vii) High expression of multi drug resistant (MDR) proteins, (viii) Microenvironment influences (ix) Presence of X-RARa fusions.

**Table 1 cancers-11-01591-t001:** Additional genetic events.

Author	Source	Number of Samples		Method	Molecular Alterations		Ref
		Dx	Relapse		Dx	Relapse	
**Madan, et al**	**Human**	163	69	WGS	FLT3 (43%), WT1 (14%), NRAS (10%) and KRAS (4%), ARID1A (5%), ARID1B (3%), LRP1 (3%)	PML(17%), RARA (10), FLT3-ITD (25%), WT1 (18%), ARID 1B (12%) RUNX1 (5%), FLT3 (5%), NRAS (5%), ARID1B(5%), NRAS (5%), ETV6 (4%), FANCA (3%), TP53 (3%), LRP1 (3%), KMT2C (3%)	[44]
Yin J, et al	Human	84	-	Genomic DNA-PCR	FLT3-ITD (27%), WT1 (14%), FLT3-TKD (10%), TET2 (8%), N-RAS (6%), ASXL1 (5%), EZH2 (2%), MLL-PTD (1%), IDH1 (1%) and CBL (1%)	-	[45]
Iaccarino, et al	Human	33	31	NGS (31-gene panel)	FLT3-ITD (34%), WT1 (20%), NRAS (7%), RUNX1 (5%), FLT3-TKD (9%), DNMT3A (5%), ETV6 (2%), MYC (2%), SETBP1 (2%), SF3B1 (5%), TET2 (%)	WT1 (13%), FLT3-ITD (10%), DNMT3A (10%), ETV6 (10%), FLT3-TKD (6%), TET2 (6%), ASXL1 (3%), JAK2 (3%), RUNX1 (3%) SRSSF2 (3%), TP53 (3%), U2AF1 (3%), PML (19%), RARa (10%)	[46]
Gaur, et al	Human	103	-	DNA Sequencing (Ex 7-8)	WT1 (4%)	-	[48]
Wartman, et al	Mouse model	-	-	NGS	Jak1 V657F or V658F and Kdm6a	-	[39]

Dx: diagnosis, Rel: relapse, WGS: whole genome sequence, NGS: next generation sequencing.

**Table 2 cancers-11-01591-t002:** Experimental strategies for resistance.

Author	Drug	Function	Study	Source	Result	Follow up	*p*	Ref
Takeshita, et al.	Tamibarotene (TAM)	RAR α agonist	Clinical trial	270 Patients	RFS %: TAM 94; ATRA 84	7- Year	0.027	[102]
Lo Coco, et al	Gentuzumab Ozogamicin	Anti CD33 + Calicamicin	Prospective Study	16 Patients Relapse	RFS % 43 ± 15%	31 month	-	[104]
Gale, et al	CEP-701 (Lestaurtinib)	FLT3 inhibitor	*In vitro*	Primary APL blast (n = 6)	Greater effect on cell survival/proliferation in FLT3/ITD cells, but this inhibition was reduced in the presence of ATRA	-	-	[105]
Mastrangelo, et al	Ascorbate Megadose	Pro-oxidant,	*In vitro*	Cells Lines (n = 6)	Highly sensitive, with an average 50 % lethal concentration (LC50) of 3 mM Normal CD34+ not sensitive	-	-	[108]
Noguera, et al	Ascorbate Megadose	Pro-oxidant,	*In vitro*	Primary APL (n = 9) and AML (n = 33) Blast; Cells Lines (n = 5)	Higer sensitivity ASC induce PML/RARa and PML degradation ASC potenciate the effect of ATO Normal CD34+ not sensitive	-	< 0.001	[107]
Masciarelli, et al	Tunicamycin	Endoplasmic reticulum (ER) stress-inducing drug	*In vitro*	Primary APL Blast; ATRA sensitive and resistant APL cell lines	ER stress + ATO induced apoptosis in RA-sensitive an RA-resistant APL cell lines	-	< 0.005	[114]
Gu, et al	pharicin B,	stabilizes RARα protein	*In vitro*	Primary APL Blast; ATRA sensitive and resistant APL cell lines	Induced apoptosis in RA-sensitive and RA-resistant APL cell lines	-	< 0.001	[115]
Wang, et al	LG-362B,	caspases-mediated degradation of PML-RARα	*In vitro* e *in vivo*	Primary APL Blast; ATRA sensitive and resistant APL cell lines Murin models	Inhibits the proliferation of APL in vitro and in vivo Synergistic or additive differentiation effect with ATRA Overcom ATRA resistance	-	RTW: < 0.01	[116]
Ying, et al	2-bromopalmitate (2-Br)	inhibitor of fatty acid oxidation	*In vitro*	Primary APL Blast; ATRA sensitive and resistant APL cell lines Murin models	ATRA + 2Br to overcoming ATRA resistance	-	Blast: < 0.05 to < 0.001 (n = 7); > 0.05 (n = 4) RTW: < 0.05	[118]
Ganesan et al	ATO plus Bortezomib	downregulation of the NFĸB pathway, PML-RARa degradation inhibition of the proteasome by bortezomib	*In vitro* e *in vivo*	ATO sensitive and resistant APL cell lines Murin models	Synergistic effect in both ATO sensitive and ATO resistant APL cell lines Reduce leukemic burden and induce long-term survival in an APL mouse model	-	OS mouse: 0.0001	[82]
Hussain et al	phenylarsine oxide (PAO)	organic arsenic derivatives	*In vitro*	Cells Lines transfected with PLZF-RARa	PLZF-RARa degradation	-	-	[119]

RFS: relapse-free survival, RTW: reduction tumor weight, OS: overal survival.

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
