# Peer review of "Acute Promyelocytic Leukemia: Update on the Mechanisms of Leukemogenesis, Resistance and on Innovative Treatment Strategies"

_cancers, 2019, doi:10.3390/cancers11101591_

Round 1

Reviewer 1 Report

This review summarize the mechanisms of leukemogenesis, therapy and resistance in APL, PML-RARa sets the cellular landscape of APL by repressing transcription of RARa target genes and disrupting PML-nuclear bodies by quoting abundant references.

Although the manuscript is well written in each section, several points should be amended or clarified for the benefit of the readers.

Minor considerations;

In APL pathophysiology, differences in APL morbidity rates among races should be described. In additional genetic events or others, the relationship between MRD and genetic traits should be described. Reportedly, the cumulative incidence of CNS relapse has been 1-2%. The prophylaxis for them is also very important. The topics for them should be described to understand deeply.

Author Response

Rome, October 4th, 1919

Dear Editor,  

We wish to thank you for the helpful revision of our manuscript “Acute Promyelocytic Leukemia: new findings about the Mechanics of Leukemogenesis, Resistance and Experimental Strategies for the Therapy”, which we have modified according to the Reviewers’ suggestions.

Detailed responses to the Reviewers’ comments follow below and they are indicated in italics. We hope that you will find our revised manuscript suitable for publication in Cancers.

Sincerely,

Nelida Ines Noguera

Specific comments:

Reviewer 1

In APL pathophysiology difference in APL morbidity among races should be described

Thank you for this suggestion. We added the following paragraph ‘Ethnicity is considered a measure of genetic and lifestyle. Some studies, mainly from Latino America, suggested that Latino population had a higher proportion of APL among all AML diagnosis, which ranged around 30-38% against less then 10-7% in the non-Latino population. Nevertheless Latino is not an ethnicity and is very difficult to define what is the characteristic of such population. On the other hand several reports did not find any significant variation in the incidence, and there is no hard evidence of any ethnic physiological or life style difference linked to APL incidence’ (page 3, lines 105-111).

In genetic events or others the relationship between MRD and genetic trait should be described.

We added the following paragraph ‘The Authors suggest that molecular patterns detectable at diagnosis may predict treatment response. Comparing the initial mutational status of patients who relapsed during the course of disease (n = 11) with those in CCR (n = 33), they found differences in concomitant mutations, with NRAS and RUNX1 mutated only in the first group of patients, suggesting their possible role as predictive markers of relapse. Of note, also the number of concomitant mutations per patient was significantly higher in the group who later relapsed (P < .0001) indicating an accumulation of genetic alterations during disease progression. In particular, they found mutations associated with clonal hematopoietic expansion, like ASXL1, DNMT3A, JAK2, SRSF2, TET2 and TP53 [47]. However, validation studies are needed to confirm these hypothesis

The optimal management of APL requires early diagnosis, institution of aggressive supportive measures, appropriate management of treatment-related complications, and monitoring of measurable residual disease (MRD) for the presence of PML/RARa. MRD monitoring is not recommended in non–high-risk patients who achieve profound molecular remission (CRMRD2) status after consolidation, independently from induction treatment (ATRA plus ATO or ATRA plus chemotherapy)[55]. Contrarily high-risk patients and those not responding well to induction therapy request frequent and accurate monitoring. As mentioned before a relatively large set of genes are mutated or aberrantly expressed in APL. These genetic alterations, over their significance in prognosis, might be of use as clonal markers in measuring MRD after treatment. The availability of new more sensitive methods of investigations, like deep sequencing and mutation-specific ddPCR, enable a better understanding of the clonal evolution in cases harbouring several mutations acquired during disease progression. But, apart from high-risk characteristics, how to discriminate at onset the cases that need a more timely diagnostic approach? The first study to propose a molecular risk score in APL was reported by Hecht et al. Although the sample size was limited (79 patients), the authors demonstrated that BAALC, ERG and WT1 expression levels integrated into a score could offer a promising approach to guide the monitoring of patients with APL treated with ATRA and high doses of cytarabine [56-57]. Recently Lucena-Araujo et al. in a recent study, propose the use of an integrative score in APL (ISAPL) based on FLT3-ITD mutational status, ΔNp73/TAp73 expression ratio, ID1, BAALC, ERG and KMT2E gene expression levels. They propose that the combination of gene mutation with gene expression analysis could improve outcome prediction in acute promyelocytic leukemia. The authors suggest that a better understanding of the molecular limitation of patients in responce to ATRA and anthracycline-based chemotherapy could serve as the basis for future initiatives aiming to change the current scenario for APL treatment in patients who live in low- and middle-income countries (LMIC) [58].’ (page 4-5 , lines 158-183 and 187-191 ).

3 reportedly the cumulative incidence of CNS relapse has been 1-2%. The profilaxis for them is also very important. The topics for them should be described to understand deeply.

In our intention the review was not meant to address specific clinical aspects, since the recent magnificent work of Sanz, Lo Coco et al seemed exhaustive for the moment of the related topics. In addition we could not find any significant breakthrough on the biology of extramedullary blasts in APL. Nonetheless, to be comprehensive we added the following paragraph: ‘Prophylaxis for incidence of CNS relapse

In the ATO era central nervous system (CNS) involvement is rare in APL but causes poor prognosis.  There are no hard biological data available concerning APL CNS localization, it had been associated with several factors including high WBC count (>10 × 109/L), CNS hemorrhage, expression of CD2, and/or CD56 in blasts, PML‐RARA bcr3 isoform, differentiation syndrome, and above all induction therapy with single agent ATRA and regimens without Cytarabine [122]. Disproved a possible association between the use of ATRA and the development of extramedullary disease [123], it is conceivable that the emergence of such a complication was more apparent due to better therapy and prolonged survival of resistant patients. ATO is known to cross the blood-brain barrier and has CNS penetration at therapeutically meaningful levels (CSF concentration at 20% to 50% of plasma concentration). Recently Sanz et al reported that there are no formal data supporting the use of CNS prophylaxis in the ATO era [124]. Also Larson et al do not recommend intrathecal chemotherapy (ITT) in the treatment of APL for any risk, thus prophylactic ITT was not used in trials that incorporated ATRA and ATO [125-128]. (page 10 , lines385-398).

Reviewer 2

For the full acceptance of the manuscript, I suggest the authors to include a table for the section “additional genetic events”. It will be easier to follow this section. In addition, I suggest another table for the last section, “experimental strategies for resistance”, as a summary of the treatments.

Thank you for this suggestion. We added Table 1 “additional genetic events”. and Table 2“experimental strategies for resistance”

Reviewer 2 Report

This is a good review related to APL mutations, their relationship with APL pathogenesis, mechanisms of resistance and current experimental treatment. This review could be really useful for researchers starting to work on APL and as an updated of current studies.

The figures are self-explanatory and useful to understand the mechanisms. 

For the full acceptance of the manuscript, I suggest the authors to include a table for the section “additional genetic events”. It will be easier to follow this section. In addition, I suggest another table for the last section, “experimental strategies for resistance”, as a summary of the treatments.

Thank you very much.

Author Response

(The authors gave the same response as above.)
